# Proteomic Analysis of Plants with Binding Immunoglobulin Protein Overexpression Reveals Mechanisms Related to Defense Against *Moniliophthora perniciosa*

**DOI:** 10.3390/plants14040503

**Published:** 2025-02-07

**Authors:** Grazielle da Mota Alcântara, Gláucia Carvalho Barbosa Silva, Irma Yuliana Mora Ocampo, Amanda Araújo Kroger, Rafaelle Souza de Oliveira, Karina Peres Gramacho, Carlos Priminho Pirovani, Fátima Cerqueira Alvim

**Affiliations:** 1Department of Biology, Santa Cruz State University, Ilhéus 45662-900, Bahia, Brazil; graziellealcantara2018@gmail.com (G.d.M.A.); glauciacbsilva@gmail.com (G.C.B.S.); yulimoraocampo@gmail.com (I.Y.M.O.); amannda152@gmail.com (A.A.K.); eng.rafaelle@gmail.com (R.S.d.O.); 2Molecular Plant Pathology Laboratory, Cocoa Research Center—CEPEC, Ilhéus 45600-970, Bahia, Brazil; gramachokp@hotmail.com; 3Proteomics Laboratory, Department of Biology, Santa Cruz State University, Ilhéus 45662-900, Bahia, Brazil; pirovani@uesc.br; 4Tissue Culture Laboratory, Department of Biology, Santa Cruz State University, Ilhéus 45662-900, Bahia, Brazil

**Keywords:** witches’ broom disease, biotic stress resistance, proteomics

## Abstract

*Moniliophthora perniciosa* is one of the main pathogens affecting cocoa, and controlling it generally involves planting resistant genotypes followed by phytosanitary pruning. The identification of plant genes related to defense mechanisms is crucial to unravel the molecular basis of plant–pathogen interactions. Among the candidate genes, BiP stands out as a molecular chaperone located in the endoplasmic reticulum that facilitates protein folding and is induced under stress conditions, such as pathogen attacks. In this study, the *Soy*BiP*D* gene was expressed in *Solanum lycopersicum* plants and the plants were challenged with *M. perniciosa*. The control plants exhibited severe symptoms of witches’ broom disease, whereas the transgenic lines showed no or mild symptoms. Gel-free proteomics revealed significant changes in the protein profile associated with BiP overexpression. Inoculated transgenic plants had a higher abundance of resistance-related proteins, such as PR2, PR3, and PR10, along with increased activity of antioxidant enzymes, including superoxide dismutase (SOD), catalase (CAT), guaiacol peroxidase, and fungal cell wall-degrading enzymes (glucanases). Additionally, transgenic plants accumulated less H_2_O_2_, indicating more efficient control of reactive oxygen species (ROS). The interaction network analysis highlighted the activation of defense-associated signaling and metabolic pathways, conferring a state of defensive readiness even in the absence of pathogens. These results demonstrate that BiP overexpression increases the abundance of defense proteins, enhances antioxidant capacity, and confers greater tolerance to biotic stress. This study demonstrates the biotechnological potential of the BiP gene for genetic engineering crops with increased resistance to economically important diseases, such as witches’ broom in cocoa.

## 1. Introduction

Cocoa (*Theobroma cacao* L.) is an agricultural crop of great economic importance worldwide as the base for the production of chocolate and a variety of other cocoa-derived products [1,2]. Global cocoa production reached around 4449 million tons in 2023, demonstrating the magnitude of this commodity. In Brazil, production reached 290,000 tons in the same year, consolidating the country as an important producer in the cocoa market [3]. However, cocoa production faces significant challenges, especially from diseases that affect cocoa trees. Among the most significant is witches’ broom disease, caused by the hemibiotrophic fungus *Moniliophthora perniciosa* [4,5], which poses a particularly serious threat, especially in the producing countries of South and Central America [6,7]. This disease, in the absence of integrated management, can cause devastating losses of up to 90% in severely affected areas. The infection causes abnormal tissue growth and deformation of shoots, flowers, and fruits, culminating in the death of the infected tissues [8,9,10]. The complex life cycle of *M. perniciosa*, characterized by monokaryotic (biotrophic) and dikaryotic (necrotrophic) phases, involves penetration into plant tissues, intercellular colonization, and the subsequent production of spores that spread the disease [4,11,12].

Responding effectively to the threat posed by witches’ broom requires an in-depth understanding of plant defense mechanisms and plant–pathogen interactions. The search for solutions to control this disease has driven research on several fronts, with emphasis placed on integrated management practices, such as crop rotation, the use of resistant varieties, and the application of appropriate agronomic techniques. These approaches are essential to reduce the spread of the pathogen and reduce damage to crops [13,14]. At the same time, research into plant defense mechanisms, especially the activation of resistance genes, is emerging as a promising alternative to increase crops’ resilience to diseases [15,16]. Proteomic analyses have excelled in identifying essential molecular targets for plant defense. [17] explored protein interactions between *T. cacao* and necrosis-inducing proteins (NEPs) from *M. perniciosa*, and identified three target proteins in cacao that interact directly with the NEP, MpNep2.

In this context, BiP (binding immunoglobulin protein) has emerged as an important protein, since its increased expression is associated with protection against cell damage [18,19]. BiP is a molecular chaperone located in the lumen of the endoplasmic reticulum (ER), belonging to the HSP70 (Heat Shock Protein) family. This class of proteins is involved in the regulation of various cellular processes, such as the translocation of proteins into the ER, protein quality control, and protection against cellular stress [20,21,22]. In the ER, BiP plays a key role in regulating the proper folding of proteins in the secretory pathway, ensuring the correct three-dimensional conformation of molecules before their secretion. During infection by *Moniliophthora perniciosa*, BiP helps control the accumulation of misfolded proteins and activates the UPR stress response. Its overexpression strengthens the plant’s defenses, increasing resistance to the pathogen by improving protein homeostasis and activating antioxidant pathways [23,24]. It acts as a stress sensor, helping to identify and degrade malformed proteins through the proteosomal degradation pathway, and is vital for maintaining protein homeostasis and preventing cell dysfunction [24,25,26]. The regulation of BiP expression is crucial for cellular homeostasis and has been identified in all genomes of eukaryotic organisms [18,27]. A signaling pathway known as the Unfolded Protein Response (UPR) is triggered when there is an increase in secretory activity or an accumulation of misfolded proteins within the endoplasmic reticulum, activated by a decrease in free BiP [23,28]. Stress events, both biotic and abiotic, lead to the activation of this pathway, where a signaling cascade, mediated by the phosphorylation of stress response initiator proteins, induces molecular chaperones including BiP (one of the most induced), to correct malformed proteins [18,23,29]. Studies indicate that increased BiP synthesis in plants is associated with a more effective response to biotic stresses, such as fungal infestations and insect attacks [30,31]. In addition, overexpression of this gene via transgenesis is directly related to an increased tolerance to abiotic stresses, such as water deficit [32,33,34].

*Soy*BiP*D*, isolated from soybean (*Glycine max* L.) [33], encodes a BiP-type chaperone protein. In a previous study, transgenic lines of *Solanum lycopersicum* var. Micro-Tom overexpressing the *soy*BiP*D* gene were tested for inoculation with the *M. perniciosa* fungus, where a correlation was observed between higher accumulation of BiP and better plant response in terms of growth, productivity, and resistance to infection. Therefore, in this study we aim to further investigate the potential of the BiP gene when overexpressed in model tomato plants (*S. lycopersicum*) in response to attack by the pathogen *M. perniciosa*. To this end, we used proteomic analysis techniques, including mass spectrometry, to identify upregulated and differentially abundant proteins (those present in both compared samples, but at different intensities) in control plants (NT) and in those overexpressing BiP, infected and not infected with the *M. perniciosa* fungus.

The results of this study can contribute significantly to the advancement of knowledge of plant–pathogen interaction, particularly in the context of resistance to witches’ broom, by providing information on the molecular mechanisms of plant defense against *M*. *perniciosa* mediated by BiP overexpression. The identification of key proteins involved in the defense response, as well as the use of the BiP gene as a target, can lead to more resistant cultivars, impacting several agricultural crops, in addition to cocoa.

## 2. Results

### 2.1. BiP Overexpression Affects Protein Composition in Plants, Under Stressed or Non-Stressed Conditions

In a previous study, we identified that transgenic *S. lycopersicum* cv. Micro-Tom overexpressing BiP presented resistance to *M. perniciosa*. Therefore, we aimed to better understand the molecular differences that could be related to our previous findings. NT and transgenic lineages were inoculated with *M. perniciosa* basidiospores, and again, we observed that NT plants showed severe symptoms of infection, including hyperplasia, overgrowth and blackening, whereas the lineages with the highest accumulation of BiP (L9, L10, and L12) remained asymptomatic throughout the experimental period (45 days after infection), with no visible signs of infection (Figure 1).

Mass spectrometry analysis was carried comparing transgenic plants (L12) and control plants (NT), both inoculated and non-inoculated with *M. perniciosa*, revealing a significant number of proteins in different experimental conditions. In non-inoculated plants, a total of 1911 proteins were identified in the NT plants, while 1909 proteins were detected in the L12 transgenic plants. In the inoculated plants, a total of 1588 proteins were identified in NT, while 1406 proteins were detected in L12. After filtering the data based on identification in 100% of the triplicates, 273 proteins were retained in the NT plants and 377 proteins in the L12 BiP plants in unstressed conditions. Considering the inoculated treatment, 254 proteins were retained in NT plants and 246 proteins in transgenic plants (Appendix A). Venn diagrams were used to show the number of upregulated and differentially abundant proteins between the experimental conditions. In the comparative non-inoculated treatment (Appendix A), 14 proteins were upregulated in the NT plants, while 34 proteins were upregulated in the transgenic plants, with 181 differentially abundant proteins between the groups. In the comparative treatment after inoculation (Appendix A), 37 proteins were identified as being upregulated in NT plants, while 30 proteins were upregulated in the transgenic plants, with 141 proteins being differentially abundant between the two groups. According to the principal component analysis (PCA) and the Venn diagram methods, all the proteins identified as being upregulated and differential were considered. However, the analysis of these proteins only included those that met the significance criteria (*p*-value < 0.05 and |fold-change| > 1.5) (Appendix A).

The heatmap shows the differentially abundant proteins for each treatment. A total of 57 upregulated and differentially abundant proteins were identified in the comparison of the non-inoculated NT plants vs. the L12 BiP plants (Figure 2A), while 78 proteins were detected as being upregulated or differentially abundant in NT vs. L12 BiP in the inoculated treatment (Figure 3A), allowing for a more detailed view of the specific changes imposed on the proteomic profile (Appendix A).

The comparative proteomic analysis of the non-inoculated heatmap (Figure 2A), revealed significant differences in specific protein abundance. Hierarchical clustering highlighted proteins that were upregulated or differentially abundant, associated with various biological processes. In the NT plants, upregulated proteins associated with energy metabolism were identified, such as glyceraldehyde-3-phosphate dehydrogenase and malate dehydrogenase, involved in glycolysis and the citric acid cycle. In the field of regulation and signaling, proteins such as histone H2A, which acts in DNA compaction, and RING-type domain-containing protein, involved in ubiquitination, stood out. In addition, the chlorophyll a-b binding protein was detected, which plays an important role in photosynthesis, especially in light capture and energy transfer. However, in the transgenic plants, upregulated proteins such as luminal-binding protein precursor (BiP), an essential chaperone for protein folding, as well as aspartic proteinase and annexin, which act in protein degradation and membrane stabilization, were identified. In energy metabolism, proteins such as triosephosphate isomerase and malate dehydrogenase were observed, while in regulation and signaling, 14-3-3 domain-containing protein and isocitrate dehydrogenase (NADP+) stood out. Among the differentially abundant proteins identified between the treatments, the most abundant proteins in the transgenic plants included those related to energy metabolism, such as glycinamide ribonucleotide synthetase, which is essential for purine biosynthesis. Proteins related to regulation and signaling were detected, such as leucine-rich repeat-containing N-terminal proteins, with potential involvement in pathogen recognition, and proteins involved in protein folding such as peptidyl-prolyl cis-trans isomerase. On the other hand, among the most abundant proteins in NT plants, carboxypeptidase, involved in protein degradation, stood out. The functional characterization of the proteins (Figure 2B) revealed that those with the greatest participation were those associated with ATPase activity and cofactor binding, both representing 26.1% of the total proteins. In addition, 21.7% of the proteins showed enzyme inhibitor activity. Proteins associated with oxidoreductive activity and heat shock protein binding were also detected, accounting for 13.0% each. In the biological processes (Figure 2B) (Appendix A), the largest proportion of proteins (21.2%) was involved in the generation of metabolic precursors and energy, followed by proteins associated with protein folding (15.2%), highlighting the role of chaperones in the endoplasmic reticulum. Other significant categories included proteins involved in glucose metabolism and the regulation of molecular functions, both accounting for 12.1%.

### 2.2. The Identification of Upregulated and Differentially Abundant Proteins Revealed That Plants That Accumulate BiP Responded Differently than Control Plants to Pathogen Infection, by Activating Defense and Oxidative Stress Proteins

In NT plants, we identified upregulated proteins related to photosynthesis and chloroplast metabolism, such as ATP synthase subunits (Δ and ε) and photosystem II oxygen evolution system proteins, such as 23 kDa subunit and PSB27-H1. We also observed metabolic proteins, such as glyceraldehyde-3-phosphate dehydrogenase (glycolysis) and ribose-5-phosphate isomerase (pentose pathway). In the field of regulation and signaling, proteins such as CP12 domain-containing protein and calmodulin-binding domain-containing protein stood out. In contrast, L12 BiP plants showed upregulated proteins related to energy metabolism and sugar metabolism, such as glyceraldehyde-3-phosphate dehydrogenase and sucrose–phosphate synthase. We also identified proteins related to regulation and signaling processes, such as the bulb-type lectin domain-containing protein, which can help regulate responses to oxidative stress and cellular homeostasis (Figure 3A).

Among the differentially abundant proteins, we observed that NT plants had a greater accumulation of proteins associated with photosynthesis, such as ATP synthase subunit beta and photosystem II stability/assembly factor, reflecting greater photosynthetic activity. Related to energy metabolism, phosphoglycerate kinase was detected, while in regulation and signaling, plastocyanin stood out, suggesting greater participation in processes related to photosynthesis and electron transport. On the other hand, L12 BiP plants showed a greater abundance of the protein related to energy metabolism, glycinamide ribonucleotide synthetase, which is essential for purine biosynthesis. With regard to regulation and signaling, peptidyl-prolyl cis-trans isomerase, involved in protein folding, was detected, indicating specific adaptations to stress and the maintenance of cellular homeostasis.

Functional characterization of the proteins (Figure 3B, Appendix A) showed that 47.1% of the proteins in the treatment were involved in tetrapyrrole binding, associated with photosynthesis and pigment metabolism. Other molecular functions included proteins with chitin-binding activities, ATP synthase activity, and specific binding to protein domains, each representing 17.6%, reflecting functional diversification in response to inoculation. In the biological processes, 16.2% of the proteins were related to the generation of metabolic precursors and energy. Proteins involved in carbohydrate metabolism and photosynthesis were also predominant, accounting for 14.4% each. In addition, there was a high participation of proteins associated with the response to biotic and abiotic stimuli (10.8%). Other processes included proteins associated with responses to chemical (8.1%) and oxidative (6.3%) stimuli, response to light (5.4%), and oxidation reduction (4.6%), showing adaptation to and defense against pathogen-induced stress.

The comparative analysis of NT and L12 BiP treatments under non-inoculated and M. perniciosa-inoculated conditions revealed differences in the abundance of proteins associated with defense and oxidative stress (Figure 4A,B). In the NT vs. L12 BiP non-inoculated treatment (Figure 4A), 16 proteins were detected. The NT plants showed the upregulation of 27 kDa acidic endochitinase, a protein associated with chitin degradation, which is a common component of fungal cell walls. In contrast, the BiP plants exhibited a more diverse protein profile, geared towards defense even without inoculation, with upregulated proteins including two isoforms of the SCP domain-containing protein, associated with pathogen recognition and triggering defense responses, and the RRL-RLP leucine-rich repeat-containing protein, known for its role in pathogen recognition. Other important defense proteins, such as the chitin-binding type-1 domain-containing protein and the bulb-type lectin domain-containing protein, were also found exclusively in L12 BiP plants, both associated with pathogen recognition and response to biotic stimuli. In addition, antioxidant proteins, such as thioredoxin domain-containing protein and annexin, were also upregulated in the transgenic plants. Although peroxidase was present in both genotypes, its accumulation was significantly higher in NT, while it was possible to detect PR10 protein, wound-induced proteinase inhibitor 1, and proteinase inhibitor II in both genotypes, with higher abundance in L12 BiP plants.

With M. perniciosa infection, we also identified diverse accumulation of defense proteins in the L12 BiP plants (Figure 4B). In the inoculated NT plants, some upregulated proteins were associated with stress and defense responses, such as the osmotin homolog, related to the response to osmotic stress, and glutaredoxi, which acts in the control of oxidative stress. In addition, the Avr9/Cf-9 rapidly elicited protein, known for its role in elicitor-activated defense, was also upregulated to NT plants. In inoculated BiP plants, there was a greater abundance of upregulated proteins, many of which are directly linked to the degradation of fungal cell wall components and defense against pathogens, such as the glucan endo-1,3-beta-d-glucosidase protein, where three isoforms were identified. Other defense and biotic stress response proteins, such as pathogenesis-related protein PR2, PR1 protein, and PR10 protein, were also only detected in the transgenic plants, reinforcing the activation of a more robust defense response. In addition, the BiP L12 plants presented upregulated antioxidant proteins, such as thioredoxin-dependent peroxiredoxin, as well as several isoforms of peroxidase with high abundance. Among the proteins shared after inoculation, superoxide dismutase stood out, which was slightly more abundant in NT plants. Although both genotypes accumulated superoxide dismutase, the NT plants were focused on a specific antioxidant response, while the L12 plants exhibited a broader profile of antioxidant enzymes.

### 2.3. Comparative Analysis of Protein–Protein Interaction Networks Revealed Distinct Clusters and Key Proteins in Plants Overexpressing BiP in Response to Stress

An analysis of the protein–protein interaction network comparing the non-inoculated NT vs. L12 BiP samples revealed a total of 1258 nodes, corresponding to proteins, and 15,670 connections, representing the interactions between them (Figure 5). Among the proteins highlighted in the interaction network, 17 were identified with greater abundance in the NT samples, while 22 proteins showed greater abundance in the L12 samples. The interactions were organized into 21 distinct clusters (CL1-CL21), each associated with different biological functionalities. The clusters with the highest number of proteins included Cluster 3 (CL3), which covers small molecule metabolic processes, with 293 proteins, and which stands out for its relevance in regulating cellular homeostasis and the production of secondary metabolites. Cluster 2 (CL2), focused on transmembrane proton transport, was the second largest, with 191 proteins. Cluster 5 (CL5), related to protein folding, also stood out, with 162 proteins. In addition to the clusters mentioned above, the analysis identified clusters associated with stress and plant defense, such as Cluster 16 (CL16), which is related to the response to heat, and Cluster 21 (CL21), which addresses the response to biotic stimuli. Further statistical analysis revealed 140 proteins classified as bottlenecks (Appendix A), of which two were detected in the NT samples. These included the homologous proteins ARG2 (arginase), belonging to CL3, and the protein EIF (iso)4G (MI domain-containing protein), grouped in CL5. In transgenic samples, six bottleneck proteins were identified, including the homologous protein Ca2 (carbonic anhydrase), which belongs to CL1, the protein A0A3Q7IIS5 (subunit B of the vacuolar proton pump) in CL2, and the proteins A0A3Q7G863 (serine hydroxymethyltransferase) and ER69 (5-methyltetrahydropteroiltriglutamate), present in CL3. Additionally, in Cluster 4, the protein A0A3Q7FZI5 (cytosolic triose triphosphate isomerase) was identified, and in Cluster 5, the protein A0A3Q7EQ38 (glycinamide ribonucleotide synthase) was observed. In addition to the proteins classified as bottlenecks, the interaction network contained 521 proteins considered hubs, which play central roles in regulating molecular interactions.

In the comparative analysis of the interaction network, the samples inoculated with *M. perniciosa* NT *vs*. L12 BiP (Figure 6) revealed a total of 1573 nodes (proteins) and 33,186 connections, which reflect the interactions between them. Among the proteins highlighted in the network, 47 were identified with greater abundance in the NT samples, while 30 proteins showed greater abundance in the L12 BiP samples. The interactions were organized into 24 distinct clusters (CL1-CL24). Among the clusters with the highest number of proteins, Cluster 2 (CL2) stood out, which is related to the generation of early metabolites and energy, containing 882 proteins. Cluster 1 (CL1), associated with the translation process, included 624 proteins, while Cluster 4 (CL4), which covers carbohydrate metabolism processes, was composed of 614 proteins.

In the interaction network of the inoculated treatment, clusters related to plant defense were identified, such as Cluster 3 (CL3), associated with cellular redox homeostasis, Cluster 16 (CL16), related to the response to biotic stimuli, and Cluster 19 (CL19), related to defense against fungi. Cluster 22 (CL22) is involved in the biosynthetic process of secondary metabolites, while Cluster 23 (CL23) is specifically classified as a defense cluster, reflecting plant responses and adaptations. In addition, the analysis revealed the presence of 228 proteins classified as bottlenecks (Appendix A), nine of which were identified as being more abundant in the NT samples. Among them, the homologous proteins CAB7-2 (chlorophyll a-b binding protein, chloroplastic), ATPb (ATP synthase subunit beta), and Ca2 (carbonic anhydrase) were grouped together in CL2. The protein A0A3Q7FFY2 (photosystem II stability/assembly factor) was found in CL3. In CL4, the proteins A0A3Q7FSA0 (pentose-5-phosphate 3-epimerase), SlFBA7 (fructose-bisphosphate aldolase), A0A3Q7HGJ9 (phosphoglycerate kinase), and AGPL3 (glucose-1-phosphate adenylyltransferase) were grouped together. In CL7, the PIIF (wound-induced proteinase inhibitor 1) protein was identified. In BiP L12 plants, four bottleneck proteins were detected, including the homologous protein Cyp-3 (cysteine proteinase 3) in CL1 and the protein A0A3Q7F9X5 (cytochrome c domain-containing protein) in CL2. The 2-CP2 (thioredoxin-dependent peroxiredoxin) protein was identified in CL3 and the PMEU1 (pectinesterase inhibitor U1) protein in CL14. The interaction network also included 661 proteins considered to be hubs.

### 2.4. Immunodetection Validated the Higher Accumulation of Defense Proteins in BiP-Transgenic Plants

Western blot analysis revealed significant differences in the accumulation of BiP, catalase (CAT), β-1,3-glucanase (PR2), and chitinase (PR3) proteins between transgenic and NT plants under both inoculated and non-inoculated conditions with M. perniciosa. The accumulation of BiP (~70 kDa) was consistently higher in the transgenic plants compared to the NT plants. The NT plants exhibited low BiP accumulation regardless of inoculation. In transgenic plants, BiP accumulation was significantly higher, with the L12-I line showing the highest accumulation after inoculation (*p* < 0.05) (Figure 7A).

For catalase (~55 kDa), the lowest accumulation was detected in inoculated NT plants (NT-I), while higher accumulation was observed in non-inoculated NT plants (NT-NI). In transgenic plants, CAT accumulation was considerably higher, particularly in the L12-I and L9-I lines, which exhibited the highest accumulation after inoculation (*p* < 0.05) (Figure 7B). The accumulation of PR2 (~25 kDa) was significantly higher in the transgenic lines L9-I, L12-NI, and L12-I. In NT plants, the highest accumulation was observed in NT-NI, followed by NT-I (*p* < 0.05) (Figure 7C). The PR3 protein (~25 kDa) was detected in all samples, with the lowest accumulation observed in NT plants. In transgenic plants, PR3 accumulation was significantly higher, particularly in the L12-NI and L12-I lines (*p* < 0.05) (Figure 7D).

### 2.5. BiP Accumulation Was Related to Lower H_2_O_2_ Content and Higher Antioxidant Enzyme Activity

NT plants inoculated with M. perniciosa showed the highest accumulation of H_2_O_2_, in comparison the inoculated transgenic lineages (*p* < 0.05). Among the transgenic lineages, L12 had the lowest accumulation of H_2_O_2_, while L2 and L9 showed intermediate levels, but lower than those of the inoculated NT plants. In the non-inoculated plants, the accumulation of H_2_O_2_ was consistently lower in all transgenic lineages (L2, L9 and L12) compared to NT plants (Figure 8A). We further analyzed hydrogen peroxide (H_2_O_2_) content using a histochemical test (DAB) confirming the quantitative data. In the non-inoculated treatment, the transgenic plants showed uniform and less pronounced staining, while the NT plants showed slightly more intense staining. After inoculation, the transgenic plants had less intense coloration compared to the NT plants, where a greater accumulation of H_2_O_2_ was observed, as evidenced by the darker brown color (Figure 8B).

We further analyzed the activity of enzymes known to be related to detoxification, such as superoxide dismutase (SOD); guaiacol peroxidase (GPX), and pathogen response, like β-1,3-glucanase.

SOD activity showed significant variations between all transgenic lineages (L2, L9 and L12) and NT plants (Figure 9A). Transgenic lineages L9 and L12 had the highest SOD activities, significantly higher than the NT plants (*p* < 0.05) in non-inoculated or inoculated treatment. L2 lineages showed intermediate activity in both inoculated and non-inoculated plants, while the NT plants had the lowest levels of SOD activity in both conditions. Inoculation with M. perniciosa intensified SOD activity in the transgenic lineages, with significant increases compared to non-inoculated plants (*p* < 0.05).

Guaiacol peroxidase (GPX) activity (Figure 9B) followed a similar pattern to that observed for SOD. The L12 lineages exhibited the highest GPX activity in inoculated plants, with a significantly higher value than NT plants (*p* < 0.05). In non-inoculated plants, lineages L9 and L12 also maintained high GPX enzyme activity, while NT plants had the lowest values (*p* < 0.05). The L2 lineage, although it showed increased levels of GPX in both conditions, had lower values than L9 and L12 plants. Inoculation resulted in a significant increase in GPX activity in transgenic plants compared to non-inoculated plants (*p* < 0.05).

β-1,3-glucanase activity (Figure 9C) was considerably higher in inoculated transgenic plants compared to inoculated NT plants (*p* < 0.05). Lineage L12 showed the highest β-1,3-glucanase activity among all those tested, followed by L9 and L2, which also showed significant increases compared to NT plants (*p* < 0.05). In non-inoculated plants, β-1,3-glucanase activity remained relatively stable and similar to the control, with no significant differences between transgenic and NT plants (*p* > 0.05).

## 3. Discussion

Overexpression of the molecular chaperone BiP in transgenic plants has been associated with their response to osmotic stress and drought tolerance by maintaining cellular homeostasis and delaying hypersensitive cell death [33,35,36,37]. Despite its association with abiotic stress, little is known about BiP’s role in plant resistance to biotic stress. Previous studies have mainly focused on metabolic [38,39] and signaling pathways [40] that suffer modifications under BiP accumulation and that could be related to the plant–pathogen interaction. In contrast, we focused on the fundamental protein composition modification imposed by BiP accumulation that could be related to plant resistance to fungal attack. For this, it was essential to first obtain transgenic *Solanun lycopersicum* plants that overexpressed BiP and had acquired resistance to the very aggressive pathogen, *M. perniciosa*.

*M. perniciosa* is a high-impact pathogen, responsible for plant death and reduced productivity. Infection by this fungus activates immune responses in plants, including the activation of defense proteins and the modulation of cellular signaling mechanisms, which are essential for controlling the infection and reducing damage [41]. The resistance observed in transgenic plants overexpressing BiP against *M. perniciosa* suggests that BiP accumulation may alter the plants’ protein composition, enhancing their defense against the fungus.

The observation that the total number of proteins identified in transgenic and NT plants was similar suggested that BiP overexpression alone did not cause significant changes in the plants’ basic metabolic processes. However, the protein composition was variable. NT plants showed a higher abundance of proteins related to energy metabolism, such as glyceraldehyde-3-phosphate dehydrogenase (GAPDH) and malate dehydrogenase (MDH), both playing a role in glycolysis, generating ATP and metabolic intermediates that are essential for cell survival. In addition, under oxidative stress, this protein can undergo oxidative post-translational modifications (oxPTMs), such as sulfenylation and nitrosylation, which redirect its function towards cell signaling. These changes allow it to act in processes such as the response to oxidative damage and the regulation of apoptosis, highlighting its relevance as a mediator of cellular responses to stress [42]. MDH, a member of the citric acid cycle, contributes to the production of NADH, which is essential for the continuity of metabolic flows and for maintaining the reducing power in cells [43]. This activity is important for NT plants, allowing them to meet normal metabolic demands and face the challenges imposed by environmental stresses. In addition, proteins related to structural and genomic regulation have been identified, such as histone H2A and the RING-type domain-containing protein, indicating a greater control over DNA compression and protein regulation by ubiquitination, fundamental processes for maintaining cellular balance [44,45]. In contrast, in transgenic plants, proteome BiP stood out, indicating the overexpression of this chaperone. In addition, the increased abundance of proteins such as triosephosphate isomerase (TPI), a crucial enzyme for glycolyses, and 14-3-3 domain-containing protein, involved in cell signaling processes, indicated adjustments in metabolism and cell signaling. Together, these results suggest that different pathways, related to cell maintenance and immune responses in plants, are triggered in BiP-transgenic plants [46,47] even in the absence of stressors. Indeed, it has been reported that the acquired tolerance to drought of transgenic plants overexpressing BiP [33] is somehow related to the UPR (unfolded protein response) pathway, which induces ER-associated quality control genes to promote the restoration of ER homeostasis [48,49,50,51,52,53]. Hence, BiP-mediated resistance to *M. perniciosa* must be linked to its capacity to modulate stress mediated cell death pathways negatively [51,54] as well as other pathways related to stress, giving the plant an advantage in responding to stressors when they appear.

After *M. perniciosa* inoculation, protein profiles revealed different strategies adopted by NT and BiP-transgenic plants. In NT plants, proteins that play key roles in the stability and functionality of photosystem II stood out, such as the ATP synthase subunits (Δ and ε) [4], the 23 kDa subunit protein (PsbP) and PSB27-H1. PsbP contributes to the efficiency of water photolysis [55], while PSB27-H1 helps repair PSII under stress [56]. Further, the presence of ribose-5-phosphate isomerase reinforces the importance of the pentose pathway, which is essential for nucleotide synthesis and cell regeneration [57], indicating the need for greater energy input to cope with biotic stress. On the other hand, when challenged with *M. perniciosa*, BiP-transgenic plants’ protein profile showed a metabolic direction aimed at energy mobilization and the maintenance of cellular homeostasis, such as glycinamide ribonucleotide synthetase [58], sucrose-phosphate synthase [59,60], and peptidyl-prolyl cis-trans isomerase [61]. These proteins are mainly related to production of nucleotides necessary for cell development, sucrose production, and to provide additional energy contributing to signaling and metabolic support under stress and protein folding, even under pathogen infection [61]. The biological process of the proteins confirmed these differences, indicating that NT plants prioritize photosynthetic processes to maintain basic energy flows, while BiP-transgenic plants adjust their energy metabolism and cell repair mechanisms.

We further analyzed specific proteins related to plant defense mechanisms, such as PRs and antioxidant enzymes. The higher abundance of defense proteins in BiP-transgenic plants suggests they have a pre-established state of defense, even in the absence of infection, which is usually characterized by the basal activation of response mechanisms to biotic and oxidative stress. In other pathosystems, this state of metabolic alert enables the early activation of proteins associated with the recognition of pathogen molecular signals, even before direct contact with the infectious agent, enabling a faster and more effective response [62,63]. In fact, it has already been shown that BiP accumulation in transgenic *Arabdopsis thaliana* and *Glycine max* promotes a delay in the perception of stress symptoms in plants and negatively regulates the process of programed cell death (PCD) [35,54]. The higher accumulation of peroxidase and wound-induced proteinase observed on NT plants in the absence of *M. perniciosa* suggests that, even in control conditions, NT plants deal with a stronger molecular stress situation in comparison to BiP-transgenic plants [64,65]. We decided to further quantify antioxidant enzymes’ activities in three different BiP-transgenic lineages (L2, L9 and L12) to evaluate if the mechanism was conserved between the lineages. The observation that all BiP-transgenic lineages tested presented a higher activity of catalase, GPX and SOD, in the absence of or during *M. perniciosa* infection, suggests that with BiP accumulation, plants have an improved ability to eliminate ROS. This mechanism must be especially important to plant defense against *M. perniciosa*, since the production of ROS, including superoxide and hydrogen peroxide, has already been reported to act as an initial barrier to limit the progression of witches’ broom disease [66,67,68]. In different pathosystems, these enzymes have been shown to play key roles in plant defense against pathogens. For example, in citrus rootstocks infected by *Phytophthora nicotianae*, the most tolerant genotypes exhibited greater regulation of CAT, SOD, and GPX, which was crucial for controlling H_2_O_2_ levels and maintaining redox homeostasis, while more sensitive genotypes showed excessive accumulation of ROS and low antioxidant efficiency [69]. In addition, the potential of guaiacol as an antioxidant agent was highlighted by its fungicidal activity against *Fusarium graminearum*, where its ability to modulate the pathogen’s oxidative metabolism reduced fungal growth and the production of deoxynivalenol (DON), reinforcing the critical role of antioxidant enzymes in reducing oxidative stress during plant–pathogen interactions [70]. The quantification of hydrogen peroxide in plants validated the proteomic data, since the NT plants had the highest levels of hydrogen peroxide, while transgenic plants contained significantly lower levels, in both inoculated and non-inoculated conditions.

Mass spectrometry also revealed that other proteins related specifically to plant defense mechanisms were more abundant on BiP-transgenic plants, even in the absence of *M. perniciosa*, such as RRL-RLP Leucine-rich repeat-containing protein, PR-10, and PR1. RRL-RLP leucine-rich repeat-containing protein is recognized in other pathosystems for its function in immunity triggered by pathogen-associated molecular patterns (PAMP-Triggered Immunity, PTI), allowing for the recognition of conserved structures, such as flagellins in bacteria and chitins in fungi [71]. Its lower abundance in NT plants may indicate a limitation in the efficiency of the immune response to infectious agents. Interestingly, the pathogenesis-related proteins PR-10 and PR-1 have already been reported to be part of a defense dynamic that occurs in cocoa plants during their interaction with the fungus *M. perniciosa* [68,70]. In witches’ broom, PR10 contributes to the inhibition of *M. perniciosa* replication and facilitates the plants’ defense response against infection [72] while PR1 is associated with the plants’ immune response, playing key roles in the recognition of pathogens and the activation of defense responses [73,74]. Therefore, the highest level of this PR protein in transgenic plants must provide a molecular advantage to face *M. perniciosa* infection, or any other stressful situation, strengthening the plants’ defensive capacity and preparing them to deal more effectively with future aggressions. When challenged with *M. perniciosa*, BiP-transgenic plants presented an even higher abundance of defense proteins such as PR1, PR 2 (glucan endo-1,3-beta-D-glucosidase), PR3 (chitin-binding type-1 domain protein) and PR10. These PR proteins may act as antimicrobial agents and modulators of the cellular environment, creating unfavorable conditions for the growth and spread of the pathogen [75]. The higher accumulation of PR protein was also confirmed when we quantified the activity of glucanase (PR2) and by the Western blot assay (PR 1 and PR10). The higher accumulation of PRs after *M. perniciosa* inoculation reinforces the role of BiP in promoting the production of defense-related proteins, which is essential for the replication and protein synthesis of the invaders [72,76]. Santos et al. (2023) revisited the molecular mechanisms of the interaction between *T. cacao* and *M. perniciosa*, highlighting the role of proteins such as PR1, PR2, PR10, peroxidase and superoxide dismutase (SOD) in the defense against the pathogen, suggesting that these proteins are promising biotechnological targets for genetic engineering programs aiming to achieve resistance to witches’ broom disease. Similarly, the present study with BiP-transgenic plants also identified key proteins associated with defense, such as PRs and antioxidants, corroborating the importance of these elements in modulating the response to oxidative stress and defense against fungal infections. These findings offer new perspectives on the use of BiP as a biotechnological tool in the development of cocoa cultivars that are more resistant to *M. perniciosa*, with implications for disease control. Further studies, including investigations into the specific molecular mechanisms involved in resistance, are needed to clarify the role of BiP in the plant–pathogen interaction and to explore its potential in other crops susceptible to fungal diseases (Figure 10). 

## 4. Materials and Methods

### 4.1. Generation of BiP-Transgenic Plants

Tomato plants (*S. lycopersicum* cv. Micro-Tom) were transformed, via *Agrobacterium tumefaciens*, with a cDNA sequence from the *soy*BiP*D* gene (GenBank accession number AF031241) [33]. Expression was controlled by the CaMV35S promoter and an enhancer of the alfalfa mosaic virus (AMV) sequence, followed by the terminator sequence of the nopaline synthase (nos) (Appendix A). The selection marker used for obtaining the transgenic plants was the resistance gene for kanamycin. Identification of the positive transgene was carried out by polymerase chain reaction (PCR) using specific primers for the *nptII* gene (neomycin phosphotransferase) (Appendix A). The transformation efficiency was 80%.

### 4.2. Plant Inoculation with M. perniciosa

Seeds were collected from BiP-transgenic plants (lineages L2, L4, L9, L10 and L12). These lines were selected according to the differential accumulation of BiP validated by the western-blot assay. All lineages presented a higher BiP accumulation in comparison to NT plants. However, L2 and L4 presented lower and L9, L10, and L12 higher BiP accumulation when comparing transgenic lines. Non-transformed plants (NT) were germinated directly in pots containing commercial (Basaplant®/compo expert). Twenty days after germination, the leaves were reduced to 1/3 of their size and the apical meristems of the seedlings were inoculated with 20 µL of a *M. perniciosa* basidiospore suspension (containing 5 × 10^5^ spores/mL of the inoculum with in vitro spore germination >80%). Basidiospores were previously produced from *M. perniciosa* isolated from *Solanum stipulaceum* (Caiçara), obtained from CEPLAC/CEPEC, Bahia, Brazil. Seven replicates of each transgenic lineage and non-transgenic plants (NT) were inoculated with the spores and the same number of plants were inoculated with agar and water to serve as a control. Plants were kept in a humid chamber (relative humidity >80% and temperature around 25 ± 1 °C). After 48 h, the plants were transferred to a greenhouse, where they remained until the end of the experiment. Plant samples for enzymatic and proteomic assays were collected at 15 (for hydrogen peroxide analyses) and 45 days (mass spectrometry, Western blot, and enzymatic analyses) after *M. perniciosa* inoculation. The experimental design was entirely randomized.

### 4.3. Proteomic Analysis

Total proteins were extracted [77] (with modifications) from three biological replicates of freeze-dried *S. lycopersicum* tomato leaves, both from non-transformed plants (NT) and from the L12 transgenic lineage plants, under controlled conditions (inoculated with agar and water) and 45 days after inoculation with *M. perniciosa* basidiospores. The period of 45 days after inoculation was selected for the analysis since in this interval, the symptoms of infection by *Moniliophthora perniciosa* were more advanced, allowing for a more precise assessment of the plants’ response. At this stage, the infection was already clearly evident, but the pathogen had not yet reached the stage of causing plant necrosis and death, which made it possible to study molecular interactions and plant defenses before the irreversible progression of the disease. The choice of lineage L12 was based on a previous study in which this line presented the highest accumulation of BiP and no witches’ broom disease symptom after *M. perniciosa* inoculation (Appendix A). Total protein extraction started with the maceration of 0.8 g of the plant material in liquid nitrogen containing polyvinylpyrrolidone (PVP). Sequential washes were carried out followed by centrifugation at 10,000 G for 15 min at 4 °C for each stage: three washes with 100% acetone, followed by three washes with 80% acetone, two washes with TCA (trichloroacetic acid) in acetone, and two washes with TCA in water. The tissue was then resuspended in 1 mL of extraction buffer containing 500 mM Tris-HCl, 100 mM KCl, and 2% (*w*/*v*) β-mercaptoethanol. Homogenization was carried out in a vortex for 15 min. Then, 1 mL of phenol was added to the samples, which were homogenized in a vortex for 10 min, followed by centrifugation at 10,000 g for 15 min at 4 °C. The supernatant was carefully transferred to a new tube and the procedure was repeated for a second clean-up with phenol. To precipitate the proteins, the contents of the tubes holding the supernatant were added to five times the volume of ammonium acetate with methanol and incubated at −20 °C overnight. The proteins were recovered by centrifugation at 10,000 g for 15 min at 4 °C and resuspended in 400 µL of 8 M urea for subsequent analysis. Protein quantification was carried out using the 2-D Quant Kit according to the manufacturer’s instructions (GE Healthcare), and different concentrations of bovine serum albumin (BSA) were used as a standard to generate a standard curve for quantifying the samples. For this purpose, 40 μg of each sample was resolved by SDS-gel (sodium dodecyl sulfate—polyacrylamide gel electrophoresis) using electrophoresis mini cubes (Hoefer) with 8 × 10 cm gels containing 12.5% acrylamide. Protein profiles were visualized after gel staining with 0.08% colloidal Coomassie (Appendix A).

The peptides from the non-transformed plant samples (NT) and the transgenic lineage (L12), both non-inoculated and inoculated with *M. perniciosa*, were digested with Tripsin and desalted using C18 resin tips (100 µL; Thermofisher^®^ Waltham, MA, USA) The peptides were eluted with 50 µL of a solution containing 75% acetonitrile, 25% water, and 0.1% formic acid, optimized to ensure maximum recovery and solubility of the peptides.

The peptides were separated with an Agilent 1290 Infinity II HPLC, system equipped with a C18 reverse phase column (AdvanceBio Peptide Mapping 2.1 × 250 mm; Agilent, Santa Clara, MA, USA) maintained at 55 °C. A 20 min elution gradient was applied using mobile phases A (H_2_O and 0.1% formic acid) and B (acetonitrile and 0.1% formic acid), with the following proportions of phase B, 5% to 35% (1–10 min), 35% to 70% (11–14 min), 70% to 100% (15–18 min), and 100% (18–20 min), followed by a 5 min column stabilization period. The samples were injected in triplicate to ensure reproducibility of the results.

Mass spectrometry analysis was carried out with an Agilent 6545 LC/QTOF spectrometer operating in Auto MS/MS acquisition mode with a selection of up to 10 precursors per cycle. The precursor selection criteria were a detection limit of 1000; 10,000 counts/spectrum; purity restricted to 100%; purity cutoff at 30%; isotopic model for peptides and charge preference for 2, 3, >3 and unknown charge. The collision energy was adjusted according to the formula (slope) × (*m*/*z*)/100 + Offset, with slope and Offset values ranging from 3.1 to 5 and from −4.8 to 10, respectively, depending on the charge of the precursor. The instrument parameters were an ionization gas temperature of 325 °C, gas flow at 13 L/min, capillary voltage at 4000 V, and skimmer voltage at 56 V. Collision-induced dissociation (CID) was carried out with nitrogen gas. System control and parameter configuration were carried out using the Agilent MassHunter Acquisition software (version 10.1/Build 10,148).

The mass spectra obtained were processed in triplicate for peptide identification using the Spectrum Mill software (Rev B.06.00.203 SP1; Agilent). The spectra were extracted using the following parameters: MS noise threshold of 10 counts; fixed carbamidomethylation modification; precursor mass range of 200 to 6000 Da; and retention time tolerance of ± 60 s; and *m*/*z* tolerance of ±1.4. The precursor charge was determined automatically. After extracting the MS/MS spectra, a search was carried out in the UniProt *S. lycopersicum* database “https://www.uniprot.org/ (accessed on 10/07/2024)”. The parameters for comparing the MS/MS spectra were maximum number of missed cleavages of 4; fixed modification of carbamidomethylation (C); variable modifications of oxidized methionine (M), pyrrolidonecarboxylic acid (N-terminal Q), deamidation (N), phosphorylation of serine (S), threonine (T), and tyrosine (Y); minimum combined peak intensity of 10%; and precursor mass tolerance of ± 10 ppm. The search results were validated and filtered, selecting only peptides with a false discovery rate (FDR) less than 1%. The final results were exported in protein–protein comparison format in an MPP APR file.

Statistical analysis to identify proteins with differential abundance was carried out using the Mass Profiler Professional 15.1 software (MPP; Agilent). The abundance of each protein was estimated by the median abundance of its identified peptides. For each treatment, a comparative experiment was carried out, where the data from triplicates of the control group were contrasted with the triplicates of the respective treated group. Initially, the proteins were filtered based on the frequency of their peptides, retaining only those identified in 100% of the triplicates in at least one of the conditions (control or treatment). The statistical significance of the difference in abundance between the groups was assessed using the unpaired T-test, with asymptotic calculation of *p*-value and correction for multiple tests using the Benjamani–Hochberg method. Proteins were considered to be differentially expressed when they had a *p*-value less than 0.05 and fold-change greater than 1.5 (in absolute value). A comparative analysis was carried out between the treatments to identify unique proteins with differential abundance. The comparisons included NT samples not inoculated with the fungus vs. non-inoculated L12 BiP samples, and inoculated NT samples vs. inoculated L12 BiP samples. The results were visualized using principal component analysis (PCA), a Venn diagram, and a heat map, generated automatically in MPP. For the PCA and Venn diagram, all identified proteins were considered. Only proteins that met the criteria for statistical significance (*p*-value < 0.05 and |fold-change| > 1.5) were included in the heat map. The parameters used to construct the heat map clustering analysis included only unique proteins and those that statistically met the criteria of *p*-value <0.05 and fold change ≥1.5. The parameters used for these analyses were normalized intensity values, Euclidean metric distance and the Ward linkage method.

## 5. Interaction Networks

The protein interaction networks were built using the STRING 12.0 software “http://www.string-db.org (accessed on 25 July 2024)”. The following parameters were applied: reference organism *S. lycopersicum*; sources of interaction evidence: co-expression, experiments, databases and co-occurrence; maximum limit of 50 interactions per protein; and confidence level of 0.7. The subgraphs generated were combined using the Cytoscape 3.10.2 software “http://www.cytoscape.org (accessed on 2 August 2024)” [78] and the network fusion tool to generate the final networks. The analysis of gene clustering by gene ontology was carried out using the BiNGO (Biological Network Gene Ontology) plugin for Cytoscape, available at http://www.cytoscape.org [79]. The degree of functional enrichment for each cluster and ontology category was assessed quantitatively (*p*-value) using a hypergeometric distribution. A correction test for multiple comparisons was applied using the false discovery rate (FDR) algorithm, implemented in the BiNGO software (Version 3.0.5). Categories of significantly enriched biological processes (*p* < 0.05 after correction by FDR) were identified and are reported here.

## 6. Western Blot

To validate the data obtained by mass spectrometry, immunodetection tests were carried out. In addition to the L12 lineage, L2 and L9 transgenic lineages were also used. For this purpose, 40 μg of leaf protein isolated from NT and the transgenic lineages, both under control conditions and 45 days after *M. perniciosa* inoculation, were separated in a 12,5% SDS-polyacrylamide gel, using a pre-colored molecular weight marker (Kaleidoscope, Bio-Rad, Hercules, CA, USA) as a reference. The proteins were transferred to nitrocellulose membranes (Bio-Rad) in transfer buffer (25 mM Tris, 0.2 M glycine, 10% methanol) for 1 h and 20 min at 250 mA. Transfer efficiency was checked by staining the membrane with Ponceau, followed by washing with Tris-buffered saline (TBS; 20 mM Tris, 150 mM NaCl, pH 7.4). Membranes were blocked with a solution of 10% skimmed milk in TBS containing 0.05% Tween-20 (TBS-T) for 12 h at room temperature. The membrane was then incubated with the specific primary antibodies for the proteins of interest: BiP (anti-BiP), catalase (anti-CAT), PR2 (anti-PR2) or Chitinase PR3 (anti-PR3), for 1 h under agitation. After washing with TBS-T, the membrane was incubated with secondary antibody conjugated to alkaline phosphatase (rabbit anti-IgG) for 1 h at room temperature. The signal was detected using the substrate solution BCIP/NBT (5-bromo-4-chloro-3-indolyl phosphate/nitro blue tetrazolium) in Tris buffer pH 9.5, until the bands were visualized. The accumulation of referred proteins was further quantified from the membrane images using the Gel Quant Net 1.8.2 software.

## 7. H_2_O_2_ Determination

Five leaf disks from NT and transgenic lineages (L2, L9, and L12), inoculated or not with *M. perniciosa*, were isolated from the third fully expanded leaf, 15 days after inoculation, and were subjected to histochemical staining with DAB-HCl (3,3′-diaminobenzidine) [80]. The leaf disks were photographed with a stereoscopic microscope (Leica EZ4) at 4× magnification. The intensity of the brown color, resulting from the reaction of DAB with H_2_O_2_, was considered indicative of the hydrogen peroxide content present in the tissue. As a negative control, leaf disks were infiltrated with a solution of water and HCl at pH 3.8 and subjected to the same washing and fixing procedure. The quantification of hydrogen peroxide (H_2_O_2_) was performed using a colorimetric assay with potassium iodide (KI), adapted from [81]. Extracts were prepared from a pool of lyophilized tomato leaves, comprising five biological replicates from non-transformed plants (NT) and transgenic lines (L2, L9, and L12), under both control conditions and after inoculation with *M. perniciosa*. The colorimetric reaction was performed with 475 µL of KI and 100 µL of 100 mM potassium phosphate buffer (pH 7.5) mixed with 25 µL of H_2_O_2_ sample or standard. The absorbance was measured at 390 nm using a microplate spectrophotometer (Spectramax Paradigm, Molecular Devices) with the SoftMax Pro 6.3 software. Data were subjected to analysis of variance (ANOVA), and treatment means were compared using the Tukey test at 5% significance (*p* < 0.05).

## 8. Enzymatic Activities

Enzyme extracts were prepared from freeze-dried Micro-Tom tomato leaves, using a pool of five biological replicates of control plants (NT) and transgenic lineages (L2, L9 and L12), both under control conditions and 45 days after stress with *M. perniciosa*. Then, 40 micrograms of freeze-dried leaf was macerated in liquid nitrogen in the presence of polyvinylpyrrolidone (PVP), and 800 µL of the extraction buffer (variable according to each enzyme) was added and samples were homogenized by sonication with a probe ultrasonicator (Gex 130, 130 W) at an amplitude of 70%, in cycles of 5 s of pulse and 10 s of break. The samples were then centrifuged at 13,400 rpm for 10 min at 4 °C. The supernatant (crude extract) was used immediately in the enzymatic assays and analyzed with a microplate spectrophotometer (Espectramax Paradigm, Molecular Devices) using the SoftMax Pro 6.3 software.

GPX (Guaiacol Peroxidase) activity was assessed spectrophotometrically following the method described by [82], involving 140 µL of the mix solution (130 µL of 50 mM sodium phosphate buffer with pH 6.0 and 10 µL of crude enzyme extract) and 140 µL of the reaction buffer solution (125 µL of 40 mM guaiacol, 50 µL of 0.06% H_2_O_2_ and 10 µL of phosphate buffer (50 mM sodium, pH 6.0, completed to 25 mL with distilled water). The absorbance was measured in quadruplicate at 470 nm, at 25 °C, for 3 min, with readings every 30 s. The consumption of guaiacol, which reflects the activity of GPX, was calculated using the linear regression equation y = 0.01890 + 1284x, obtained from a standard curve of peroxidase (POD) with guaiacol. The results were expressed in mmol h^−1^ g^−1^ of dry matter (DM).

SOD (superoxide dismutase) activity was also determined spectrophotometrically following the method described by [83] (with modifications). The mix was made with 50 mM potassium phosphate buffer, pH 7.8, containing 1 mM EDTA and 130 mM methionine). Then, 50 µL of the crude enzyme extract was added to the microplate. The reaction was initiated by adding 20 µL of 750 mM Nitrobluetetrazolium (NBT) and 20 µL of 1 mM riboflavin. The absorbance was measured in quadruplicate at 560 nm and 25 °C. The first reading was taken after 5 min of incubation in the dark, followed by additional readings at 10, 15, and 20 min of incubation under 20 W fluorescent light. SOD activity was expressed as the amount of enzyme required to inhibit 50% of the photochemically induced NBT reduction. Each sample was analyzed in quadruplicate.

The activity of β-1,3-glucanase was determined spectrophotometrically with an adapted version of the method described by [84]. A total of 400 µL of 50 mM sodium acetate buffer (pH 5.0), 200 µL of enzyme extract, and 200 µL of CM-Curdlan-RBB substrate were added. The resulting mix was incubated at 37 °C in a water bath for 2 h. The reaction was stopped by adding 200 µL of 2N HCl, followed by cooling in an ice bath for 10 min. The absorbance of the supernatant was measured in triplicate at 600 nm in a spectrophotometer. The activity of β-1,3-glucanase was expressed in units per milligram of protein (U/mg), where one unit corresponds to the amount of enzyme that releases 1 µmol of glucose equivalent per minute, under the assay conditions. A standard curve was constructed with glucose to quantify the reducing sugars released in the reaction.

## 9. Experimental Design and Statistical Analysis

The experiment was conducted using a completely randomized design. The data were submitted to analysis of variance (ANOVA), and the means of treatments were compared using the Tukey test at 5% significance (*p* < 0.05).

## 10. Conclusions

In this study, we found that overexpression of the *soy*BiP*D* gene in tomato plants (*Solanum lycopersicum*) conferred greater resistance to the pathogen *Moniliophthora perniciosa*. While the NT plants succumbed to the infection after showing severe symptoms, the transgenic strains L9, L10, and L12 remained asymptomatic throughout the experiment. This difference was strongly associated with the greater accumulation of BiP, which contributed to cellular homeostasis by reorganizing damaged proteins and modulating defense pathways. Our results highlight the potential of BiP as a molecular chaperone capable of increasing plant resilience, especially in environments prone to stress caused by pathogens. Proteomic analyses showed that even under normal conditions, overexpression of BiP gave the transgenic plants a pre-activated defense state, preparing them to respond more efficiently to future stresses. In response to inoculation, the transgenic plants prioritized the production of defense proteins, such as PR2, PR3, and PR10, which play key roles in the plant’s immune response. Another highlight was the superior efficiency of the L12 BiP plants in regulating the accumulation of ROS, which is essential for preventing oxidative damage and maintaining the redox balance. Antioxidant proteins such as peroxidases and PRX, along with SOD, GPX, and catalase, showed a more robust antioxidant system in the transgenic plants. In contrast, NT plants showed high levels of hydrogen peroxide, indicating a lower efficiency in controlling ROS. The broader implications of this study reinforce the biotechnological potential of the BiP gene for innovation in agriculture. These results demonstrate that the combination of enhanced basal defense, adjusted metabolic responses and efficient antioxidant systems may offer a promising strategy to face biotic challenges in species of agricultural interest, such as cocoa. This knowledge can serve as a basis for future research aimed at integrating BiP overexpression into breeding programs or genetic engineering approaches, broadening the prospects for sustainable disease management.

## Figures and Tables

**Figure 1 plants-14-00503-f001:**
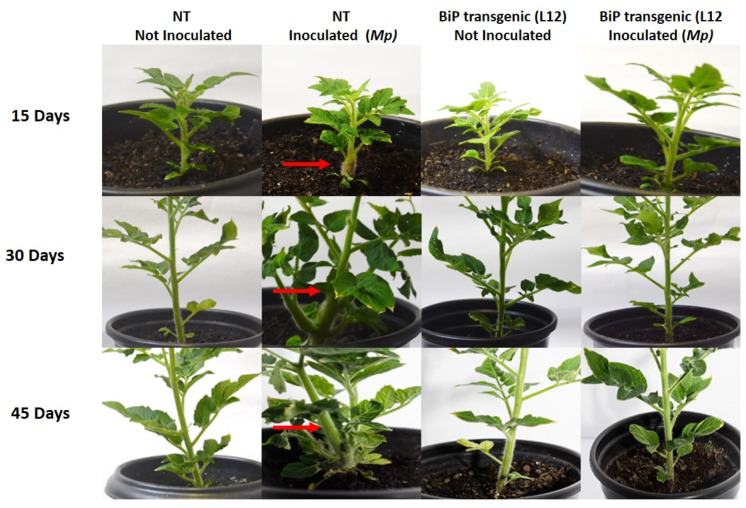
Witches’ broom symptoms in *Solanun lycopersicum* inoculated with *Moniliophthora perniciosa* (*Mp*). Inoculated NT plants developed typical disease symptoms, such as hyperplasia and overgrowth (red arrows). In contrast, inoculated transgenic plants (BiP L12) showed no symptoms throughout the observation period. Non-inoculated plants (NT and BiP L12) maintained a healthy phenotype throughout the experiment. Pictures taken 15, 30, and 45 days after inoculation.

**Figure 2 plants-14-00503-f002:**
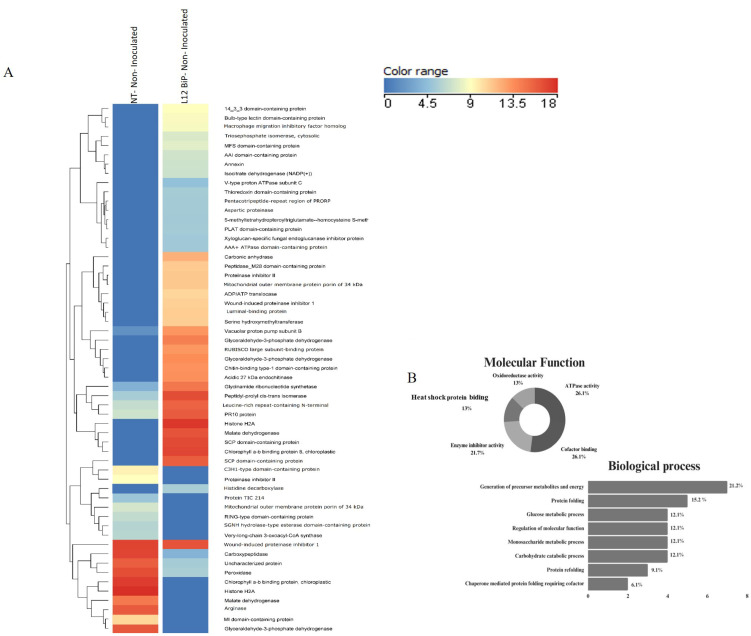
Protein identification: NT—non-inoculated vs. L12 BiP-non-inoculated treatment (*p*-value < 0.05 and |fold-change| > 1.5)—(**A**) Heatmap of *Solanum lycopersicum* leaf protein abundance (log _10_). Indicated by the scale in the figure: proteins (rows) and treatments (columns). The dendrogram shows proteins grouped according to the Euclidean distance. (**B**) Molecular function and biological process. NT, non-transformed plant line; L12 BiP, plant line overexpressing BiP.

**Figure 3 plants-14-00503-f003:**
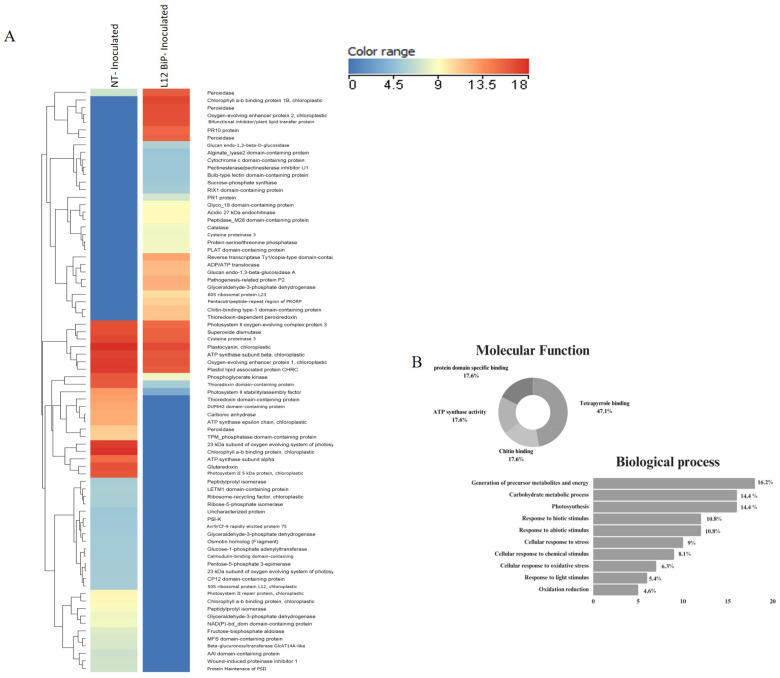
Protein identification: NT-inoculated vs. L12 BiP-inoculated treatment (*p*-value < 0.05 and |fold-change| > 1.5)—(**A**) Heatmap of *Solanum lycopersicum* leaf protein abundance (log_10_). Indicated by the scale in the figure: proteins (rows) and treatments (columns). The dendrogram shows proteins grouped according to the Euclidean distance. (**B**) Molecular function and biological process.

**Figure 4 plants-14-00503-f004:**
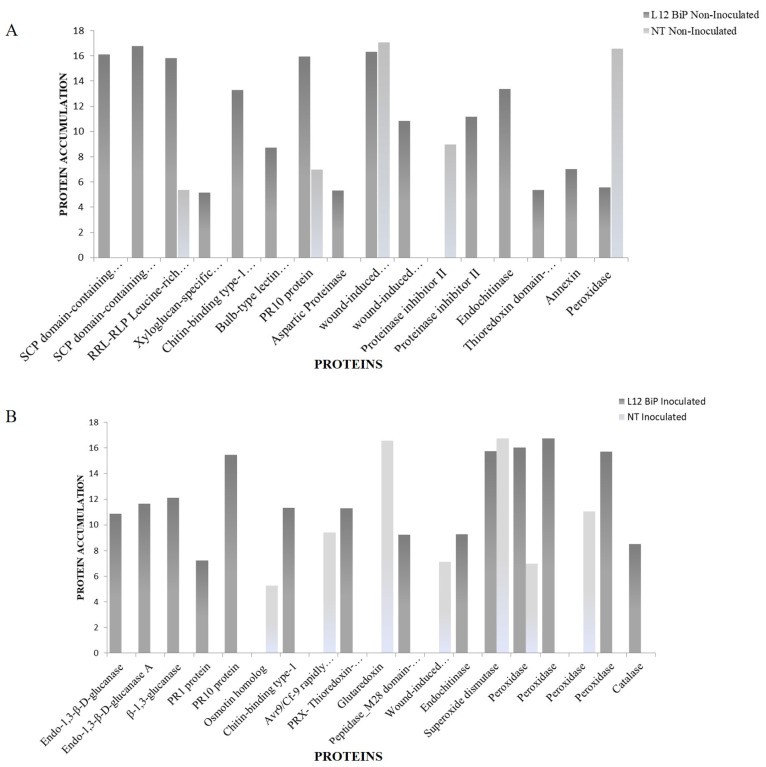
Abundance of identified proteins known to be involved in defense and stress response. In (A), a comparison of protein abundance identified in NT x L12 BiP plants not inoculated with *M. perniciosa*. In (B), a comparison of protein abundance identified in NT x vs. L12 BiP plants inoculated with *M. perniciosa*.

**Figure 5 plants-14-00503-f005:**
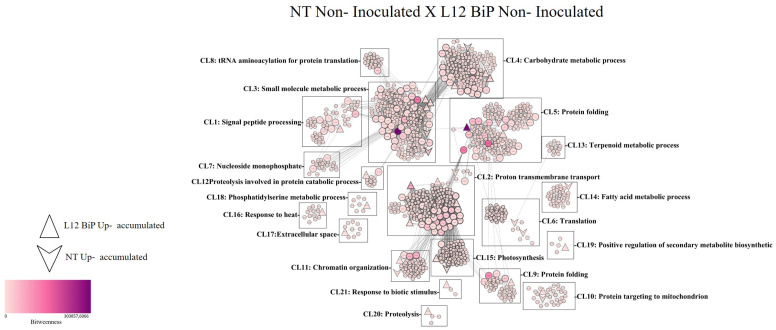
Protein–protein interaction network identified in the *Solanum lycopersicum* leaf samples. Non-inoculated NT treatment vs. non-inoculated L12 BiP. (CL1–CL 21) clusters. The betweenness value is represented by the fill color of the nodes, where the lighter color represents the lowest value and the darker color the highest value. The node degree parameter is represented by the edge width of the nodes, where nodes with a thinner edge have a lower node degree and nodes with a wider edge have a higher node degree value.

**Figure 6 plants-14-00503-f006:**
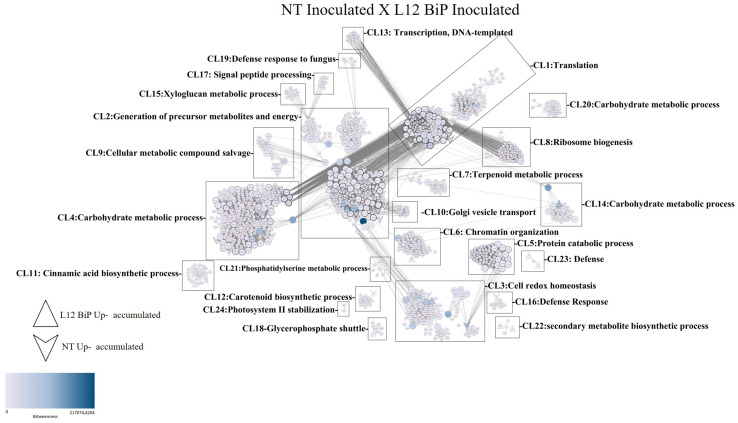
Protein–protein interaction network identified from *Solanum lycopersicum* leaf samples. Treatment NT inoculated vs. L12 BiP inoculated. (CL1–CL24) clusters. The betweenness value is represented by the fill color of the nodes, where the lighter color represents the lowest value and the darker color the highest value. The node degree parameter is represented by the edge width of the nodes, where nodes with a thin edge have a lower node degree and nodes with a wide edge have a higher node degree value.

**Figure 7 plants-14-00503-f007:**
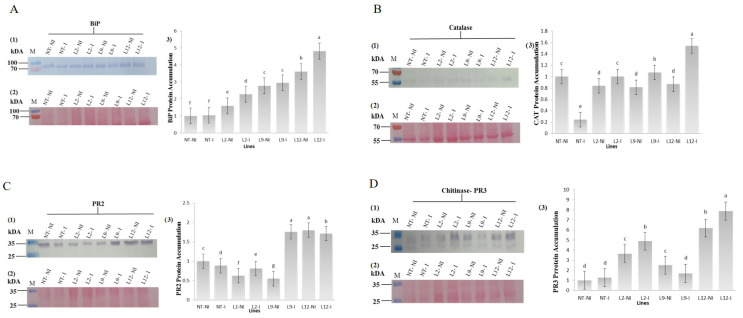
Immunodetection. (**A**) anti-BiP, (**B**) anti-catalase, (**C**) anti-PR2 and (**D**) anti-PR3 in non-transformed (NT) and transgenic (L2, L9 and L12) *Solanum lycopersicum* lines, non-inoculated and inoculated (NT I, L2 I, L9 I and L12 I) with *M. perniciosa*. (kDa) corresponds to the molecular mass (M) molecular marker; (1) protein accumulation; (2) mirror gel; (3) quantification of protein accumulation of the samples estimated through the Gel Quant. NET v1.8 program. Letters indicate significant differences between treatments (*p* < 0.05).

**Figure 8 plants-14-00503-f008:**
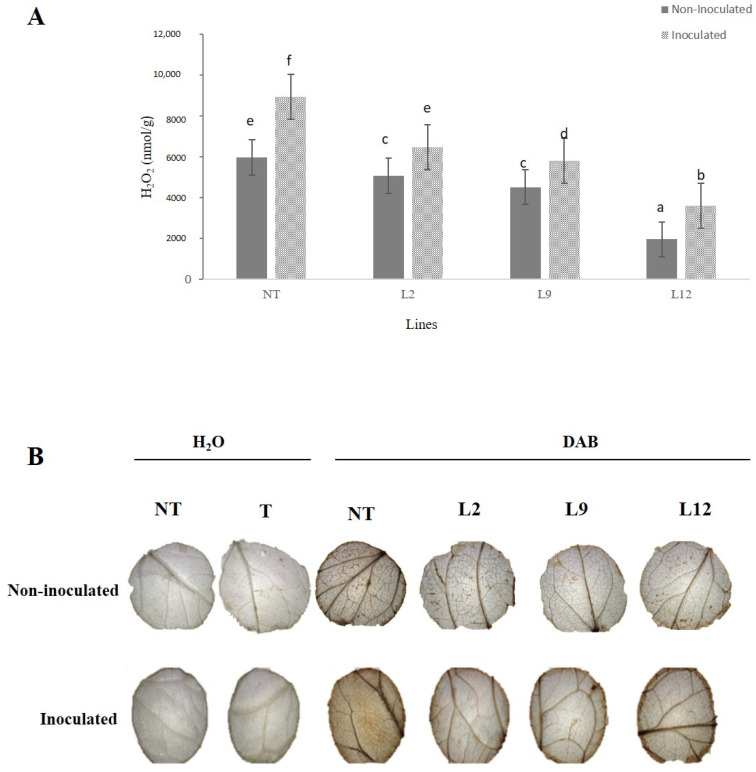
Analysis of H_2_O_2_ production and peroxide activity in tomato leaves. NT (non-transformed) (T) trangenics plants and transgenic lines (L2, L9, and L12), with or without inoculation with Moniliophthora perniciosa. (**A**) Quantification of H_2_O_2_ (mmol/g) in leaves of different tomato lines, with or without inoculation. Bars represent the mean ± standard error, and letters indicate significant differences between treatments (*p* < 0.05). (**B**) Images of leaves treated with DAB (3,3′-diaminobenzidine), showing peroxide staining in response to inoculation with M. perniciosa (inoculated) and the control (non-inoculated). The leaves were treated with either H_2_O or DAB as indicated.

**Figure 9 plants-14-00503-f009:**
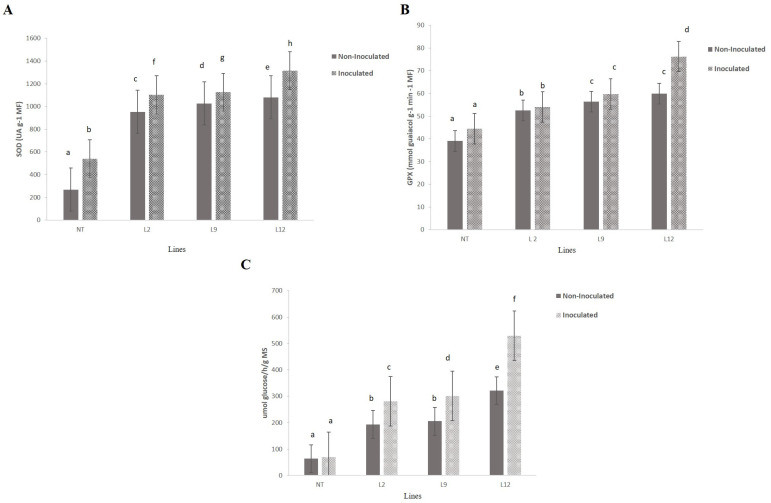
Enzymatic activity analysis of tomato leaves. NT (non-transformed) and transgenic lines (L2, L9, and L12), with or without inoculation with Moniliophthora perniciosa. (**A**) Superoxide dismutase (SOD) activity (UA at 1 mM) in different tomato lines, with or without inoculation. Bars represent the mean ± standard error, and letters indicate significant differences between treatments (*p* < 0.05). (**B**) Guaiacol peroxidase (GPX) activity (nmol glutathione·g^−1^·min^−1^·MF) in the same lines, with or without inoculation. (**C**) β-1,3-glucanase activity (µmol glucose/g MS) in leaves of the different lines, with or without inoculation.

**Figure 10 plants-14-00503-f010:**
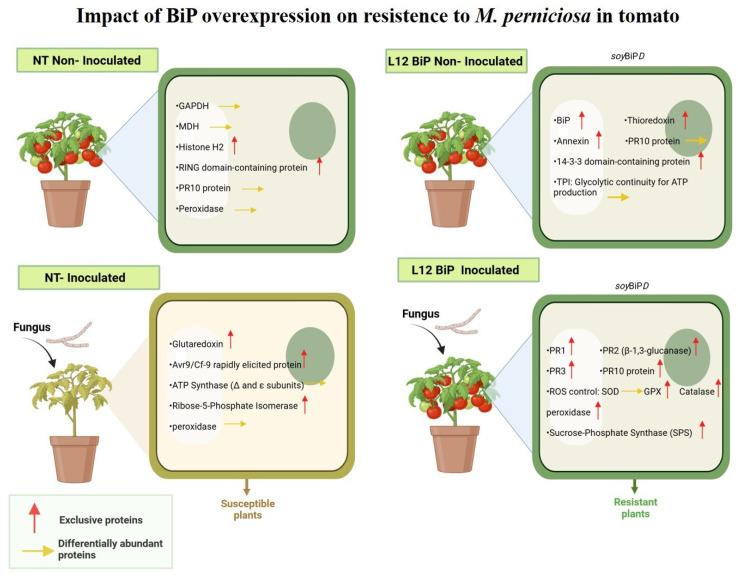
Biological model of the impact of BiP gene overexpression on the proteome of tomato plants (*S. lycopersicum*) under non-inoculated and *M. perniciosa*-inoculated conditions. NT (non-transformed) and transgenic lines (L12BiP), with and without inoculation with *Moniliophthora perniciosa*. (Red arrow) corresponds to proteins exclusive to the treatment, (Yellow arrow) corresponds to differentially abundant proteins between treatments). (NT without inoculation: General condition: basal activity focused on basic metabolic functioning. (L12BiP without inoculation). General condition: enhanced pre-established defense with metabolic and redox readiness, providing greater initial response capacity. (NT inoculated): General condition: dependence on photosynthetic processes with limited oxidative and local defensive responses. (L12BiP inoculated): General condition: robust systemic response integrated with the activation of multiple defense mechanisms, offering greater efficiency in combating the pathogen and protecting plant tissues.

## Data Availability

The data presented in this study are available in the article and Appendix A.

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
