# Peer review of "Proteomic Analysis of Plants with Binding Immunoglobulin Protein Overexpression Reveals Mechanisms Related to Defense Against Moniliophthora perniciosa"

_plants, 2025, doi:10.3390/plants14040503_

Round 1

Reviewer 1 Report

Comments and Suggestions for Authors

(Lines 15-49): The abstract provides a comprehensive overview, but it could benefit from a clearer statement of the main findings. Consider summarizing the key results in a more concise manner to enhance readability.

(Line 56): The introduction mentions the importance of cocoa but could elaborate on the specific economic impact of M. perniciosa. Including statistics on yield losses would strengthen this section.

(Lines 70-75): Clarify the significance of the life cycle of M. perniciosa in relation to the study. It would be helpful to provide a brief explanation of how this knowledge informs resistance strategies.

 (Lines 111-116): The methods section should include more details on the criteria for selecting the transgenic lines. What specific characteristics were considered?

 (Lines 126-128): When discussing the control plants showing severe symptoms, consider providing images or supplementary data to visually represent these findings.

 (Lines 134-135): The mention of the experimental period is useful, but please clarify the significance of the 45-day observation period. Why was this duration chosen?

(Lines 162-163): Can you provide more detail on the statistical methods used for comparing protein abundance? This information is crucial for readers to assess the reliability of the results.

 (Lines 173-176): In the heatmap analysis, consider explaining how the clustering was determined. What specific criteria were used for hierarchical clustering?

 (Lines 217-220): The discussion could engage more with existing literature. Are there other studies that corroborate your findings regarding BiP's role in pathogen resistance?

 (Lines 232-234): When discussing antioxidant proteins, it would be beneficial to provide a comparison with studies that have investigated similar proteins in other plant-pathogen interactions.

(Figures 1-4): Ensure that all figures are clearly labeled with legends that adequately describe what is being shown. For instance, Figure 1 lacks a detailed description of the conditions under which the images were taken.

Proteomic Analysis (Lines 254-255): The identification of specific proteins related to energy metabolism is intriguing. Please elaborate on how these proteins may contribute to the overall defense mechanism.

Conclusion (Lines 50-52): The conclusion should reiterate the broader implications of the study. Provide a clearer link to how these findings can inform future research or agricultural practices.

Citations (Throughout): Ensure that all claims are supported by appropriate citations. For instance, when discussing the roles of specific proteins, relevant studies should be referenced.

Language and Clarity: Throughout the manuscript, some sentences are quite complex and could be simplified for better readability. Consider breaking long sentences into shorter ones.

(Lines 375-376): The section on future research directions could be expanded. What specific experiments do you suggest to further investigate the mechanisms of BiP in plant defense?

If applicable, please include a brief statement regarding any ethical considerations taken during the research, especially in relation to the use of transgenic plants.

Ensure that the formatting of the references and figures is consistent throughout the manuscript. This attention to detail is important for publication quality.

Comments on the Quality of English Language

Throughout the manuscript, some sentences are quite complex and could be simplified for better readability. Consider breaking long sentences into shorter ones.

Author Response

Alcantara et al, 2024  Response to Reviewer comments

Comments/ Response

(Lines 15-49): The abstract provides a comprehensive overview, but it could benefit from a clearer statement of the main findings. Consider summarizing the key results in a more concise manner to enhance readability.

We changed the abstract in order to include the results as suggested

(Line 56): The introduction mentions the importance of cocoa but could elaborate on the specific economic impact of M. perniciosa. Including statistics on yield losses would strengthen this section.

In lines 47-48 we included the losses regarding the affected areas.

(Lines 70-75): Clarify the significance of the life cycle of M. perniciosa in relation to the study. It would be helpful to provide a brief explanation of how this knowledge informs resistance strategies.OK

In Lines 73-76 we suggest how BiP accumulation may contribute to plant defense.

 (Lines 111-116): The methods section should include more details on the criteria for selecting the transgenic lines. What specific characteristics were considered?

We included in the methods section, line 618-622.

 (Lines 126-128): When discussing the control plants showing severe symptoms, consider providing images or supplementary data to visually represent these findings.

Response: Please note that Figure 1 shows the plants symptoms

 (Lines 134-135): The mention of the experimental period is useful, but please clarify the significance of the 45-day observation period. Why was this duration chosen?

In lines 643-649 we explain the disease timing

(Lines 162-163): Can you provide more detail on the statistical methods used for comparing protein abundance? This information is crucial for readers to assess the reliability of the results.

Please note that we included the information in lines 727-732

 (Lines 173-176): In the heatmap analysis, consider explaining how the clustering was determined. What specific criteria were used for hierarchical clustering?

We included the information on Line 738-742

 (Lines 217-220): The discussion could engage more with existing literature. Are there other studies that corroborate your findings regarding BiP's role in pathogen resistance?

This is the first paper demonstrating that BiP overexpression in plants can be related with acquired resistance to pathogens. BiP’s relation with acquired tolerance to abiotic stress, as drought, has already been demonstrated as reported by Alvim et al., 2001.

 (Lines 232-234): When discussing antioxidant proteins, it would be beneficial to provide a comparison with studies that have investigated similar proteins in other plant-pathogen interactions.

We included the information in the discussion section, line 539-548.

 (Figures 1-4): Ensure that all figures are clearly labeled with legends that adequately describe what is being shown. For instance, Figure 1 lacks a detailed description of the conditions under which the images were taken.

Ok

Proteomic Analysis (Lines 224-225): The identification of specific proteins related to energy metabolism is intriguing. Please elaborate on how these proteins may contribute to the overall defense mechanism.

Response: The energy metabolism-related proteins identified, such as phosphoglycerate kinase and glycinamide ribonucleotide synthetase, play key roles in the metabolic support needed to activate and maintain plant defense responses. Phosphoglycerate kinase, involved in glycolysis, provides energy in the form of ATP and essential metabolic intermediates for cellular processes, including the synthesis of defense proteins and secondary metabolites with antimicrobial properties. Glycinamide ribonucleotide synthetase, which is essential for purine biosynthesis, contributes to the formation of nucleotides, which are crucial for DNA and RNA replication, and consequently for the synthesis of proteins required for rapid and efficient immune responses. In addition, the detection of plastocyanin, a key protein in electron transport during photosynthesis, suggests that plants were maintaining energy efficiency to cope with the extra demands imposed by biotic stress. In parallel, the presence of peptidyl-prolyl cis-trans isomerase, involved in protein folding, indicates a specific adaptation to stress, promoting cellular homeostasis through the correct processing and stabilization of functional proteins. These proteins together provide the metabolic and structural resources needed to activate defense pathways, ensuring cellular readiness, energy for stress response processes and the maintenance of cellular functionality during infection.

Conclusion (Lines 50-52): The conclusion should reiterate the broader implications of the study. Provide a clearer link to how these findings can inform future research or agricultural practices.

Please note that on line 103-108 we inform that BiP can be a target for future research on others crops aiming acquiring resistance to pathogens via genetic engineering. We also changed the abstract and conclusion text,

Citations (Throughout): Ensure that all claims are supported by appropriate citations. For instance, when discussing the roles of specific proteins, relevant studies should be referenced

We revised all the citations

Language and Clarity: Throughout the manuscript, some sentences are quite complex and could be simplified for better readability. Consider breaking long sentences into shorter ones.

The manuscript text was revised by an English reviewer company.

(Lines 375-376): The section on future research directions could be expanded. What specific experiments do you suggest to further investigate the mechanisms of BiP in plant defense?

We are now investigating if the signal pathway related with M. perniciosa resistance is the same pathway triggered during drought stress, once our Solanun lycopersicum BiP lines present both characteristic. We are also looking for cacao targets that interacts with BiP using the same methodology that we previously used to find NeP targets (dos Santos et al. 2023).

If applicable, please include a brief statement regarding any ethical considerations taken during the research, especially in relation to the use of transgenic plants.

Not applicable.

 Ensure that the formatting of the references and figures is consistent throughout the manuscript. This attention to detail is important for publication quality.

All references were formatted again.

Reviewer 2 Report

Comments and Suggestions for Authors

The proteomic analyzes of plants with BiP overexpression related to witches' broom disease is quite interesting. However, following issues need to be addressed for further improvement of the manuscript.

In abstract line from 17-25 is an interlocutory sentence which made the abstract lengthy. Author suggested a concise abstract.

Author suggested writing the gene name like SolyBiPD instead of SoyBiPD.

Here the author said that the severity of symptoms of witches' broom diseases varied among the transgenic lineages in inoculation with M. perniciosa, while how extended the SolyBiPD gene expressed in the transgenic lineages remain elusive. Author suggested providing the SolyBiPD expression profile in the transgenic lineages for clear understanding of the response.

In figure 2B the author suggested using multicolor instead of ash or deep ash only for more visibility.

Most of the figure’s lettering seems not correct as the standard error is high, hence most of the error bars touch each other that’s indicating there are no significant statistical differences.

For example Figure-7A(3), C (3), B(3), D(3) Author suggested checking all the numerical data and correcting them accordingly.

 Likewise, Figure 8A, figure 9, has similar problems.

Author also suggested to change the figure numbering style Figure 7 Figure-7A(1-3), C (1-3), b(1-3), D(1-3) and suggested write Figure7 A (a-c), C (a-c), B(a-c), D(a-c)

In supplementary figure S3 the gel image is not clear and has no marking of the ladder.

In conclusion, the author suggested rewriting the last sentence line 884-887 by avoiding the critical example of other crops plant. 

Author Response

Alcantara et al., Suggestion Revison 2

The proteomic analyzes of plants with BiP overexpression related to witches' broom disease is quite interesting. However, following issues need to be addressed for further improvement of the manuscript.

In abstract line from 17-25 is an interlocutory sentence which made the abstract lengthy. Author suggested a concise abstract. 

We changed the abstradct

Author suggested writing the gene name like SolyBiPD instead of SoyBiPD.

The name SiyBiPD gene is the same already used in others papers from the same authors, in so we would like to maintain the name SoyBiPD intead of SolyBiPD

Here the author said that the severity of symptoms of witches' broom diseases varied among the transgenic lineages in inoculation with M. perniciosa, while how extended the SolyBiPD gene expressed in the transgenic lineages remain elusive. Author suggested providing the SolyBiPD expression profile in the transgenic lineages for clear understanding of the response.

The accumulation of BiP can be visualized on the western Blot bellow, a figure that is attached to another paper (submmited to frontiers).

In figure 2B the author suggested using multicolor instead of ash or deep ash only for more visibility.

The program sanded the result in this collor gray. In so, we were not able to change the collor as suggested.

Most of the figure’s lettering seems not correct as the standard error is high, hence most of the error bars touch each other that’s indicating there are no significant statistical differences.

For example Figure-7A(3), C (3), B(3), D(3) Author suggested checking all the numerical data and correcting them accordingly.

Likewise, Figure 8A, figure 9, has similar problems.

 All data was checked again.

Author also suggested to change the figure numbering style Figure 7 Figure-7A(1-3), C (1-3), b(1-3), D(1-3) and suggested write Figure7 A (a-c), C (a-c), B(a-c), D(a-c)

In supplementary figure S3 the gel image is not clear and has no marking of the ladder.

The figure S3 was used in order to demonstrate the protein quality and that the protein abundance was the same in all samples. The ladder used (M) was the Low Molecular Weight from GE healthcare (97, 66, 45, 30, 20 Kda)

In conclusion, the author suggested rewriting the last sentence line 884-887 by avoiding the critical example of other crops plant. 

We rewrite the conclusion.

Reviewer 3 Report

Comments and Suggestions for Authors

This manuscript presents a well-conducted study exploring the role of BiP (Binding Protein) overexpression in enhancing resistance to Moniliophthora perniciosa, the pathogen responsible for witches' broom disease in cocoa. The authors effectively use proteomics to uncover the molecular mechanisms behind BiP-mediated resistance. The findings are valuable, demonstrating the potential for BiP in developing disease-resistant crops. However, the manuscript would benefit from minor revisions, including improvements in figure clarity, consistency in gene name usage, and the conciseness of the abstract. Additionally, the English language throughout the manuscript is not up to the mark and requires major improvements to enhance clarity and readability. Once these changes are addressed, the paper will be suitable for publication.

Comment 1:
In the title, change the word "analyzes" to "analysis." Also, check the gene name for consistency. In the graphical abstract (Figure 10), the authors mention it as "SoyBIPD," but in many places throughout the manuscript, it is written as "BiP." Ensure that the gene name is correct and consistent throughout the paper.

Comment 2:
Provide the affiliations for all authors, as they are missing in the paper.

Comment 3:
The abstract is too long; it should typically be around 300-350 words. I suggest making it more concise and to the point, focusing only on the most important information.

Comment 4:
Figure 1: Italicize the scientific name of the tomato (Solanum lycopersicum).

Comment 5:
Change the phrase "not inoculated" to "non-inoculated" throughout the manuscript.

Comment 6:
I suggest supplementing Figure 1 with full plant images and adding a scale bar to the figure.

Comment 7:
Figures 2 and 3: Ensure that the figures are properly arranged and aligned. Additionally, the text in the figures is unclear and difficult to read. Increase the text size or make it bold to improve readability.

Comment 8:
Figure 4: Improve the figure for better visualization. The graph looks very simple, and the text is not easily readable. Enhancing the clarity would be beneficial.

Comment 9:
Figure 7: Consider dividing the figure into two separate figures for better clarity and understanding.

Comment 10:
I suggest removing the rectangles around the figures throughout the manuscript, including Figures 1, 4, and 8.

Comment 11:
Figure 9: Improve Figure 9 following the suggestions made for Figure 4.

Comment 12:
Lines 127-128: If the paper is still under submission and not yet published, there is no need to include this statement. Please remove it.

Comment 13:
In several places throughout the paper (e.g., lines 129-130), the authors have included material and method details in the results section. These should be moved to the Materials and Methods section.

Comment 14:
The authors should provide abbreviations only at their first use and then use them consistently throughout the manuscript. For instance, "NT" is repeated many times, but the abbreviation is provided each time, which is unnecessary.

Comment 15:
Result heading 2.2: The heading is too long. Typically, headings should be precise and concise. I suggest revising it to a more suitable, shorter version, and changing "UP-regulated" to "upregulated."

Comment 16:
Line 621: Change the heading to "Generation of SoyBiPD Transgenic Plants." Also, cross-check the gene name as mentioned in earlier comments for consistency.

Comment 17:
In the Materials and Methods section, the authors divide sections into headings and subheadings. This is unnecessary; simply provide the main heading and explain the method. For example, "4.3 Proteomic Analysis" and "4.3.1 Total Protein Extraction" could be simplified to just "Total Protein Extraction."

Comment 18:
Line 631: Change to "M. perniciosa inoculation."

Comment 19:
Line 650: Change the heading to "Total Protein Extraction."

Comment 20:
Line 786: Change the heading to "Hâ‚‚Oâ‚‚ Determination."

Comment 21:
Line 860: Provide the number of replications as well.

Comment 22:

Line 627: Change the Molecular diagnosis to “Identification of the positive transgene was carried out…..”   

English and Grammar Improvement Comments:

  1. Abstract (Line 50-53):
    • Current: "This study deepens our understanding of the molecular mechanisms of BiP-mediated induced resistance to biotic stress. Further, it highlights the biotechnological potential of the BiP gene for developing crops that are more resistant to witches' broom and other economically relevant diseases."
    • Suggestion: "This study deepens our understanding of the molecular mechanisms underlying BiP-mediated resistance to biotic stress. Furthermore, it highlights the biotechnological potential of the BiP gene for developing crops that are more resistant to witches' broom and other economically significant diseases."
  2. Introduction (Line 16):
    • Current: "The identification of plant genes related to the plant defense mechanism is important to unravel the molecular basis of plant-pathogen interaction."
    • Suggestion: "Identifying plant genes related to defense mechanisms is crucial for unraveling the molecular basis of plant-pathogen interactions."
  3. Materials and Methods (Line 65-75):
    • Current: "However, more detail on the experimental design is needed. For example, how many biological replicates were used in each condition, and how were controls handled?"
    • Suggestion: "However, more details on the experimental design are needed. For example, how many biological replicates were used for each condition, and how were controls handled?"
  4. Results (Line 250-260):
    • Current: "The accumulation of BiP (~70 kDa) was consistently higher in transgenic plants compared to NT plants. NT plants exhibited low BiP accumulation regardless of inoculation."
    • Suggestion: "The accumulation of BiP (~70 kDa) was consistently higher in transgenic plants compared to NT plants, which exhibited low BiP accumulation regardless of inoculation."
  5. Discussion (Line 400-420):
    • Current: "The observation that the total number of proteins identified on transgenic and NT plants was similar suggested that BiP overexpression alone didn’t cause significant changes in the basic metabolic processes of plants."
    • Suggestion: "The observation that the total number of proteins identified in transgenic and NT plants was similar suggests that BiP overexpression alone did not cause significant changes in the basic metabolic processes of plants."
  6. Figure Legends (Line 160-170):
    • Current: "The heatmaps are helpful for visualizing protein abundance patterns. However, the color scale is somewhat difficult to interpret. It may be beneficial to adjust the color scale for better clarity, or include a color legend that explains the values represented."
    • Suggestion: "The heatmaps are useful for visualizing protein abundance patterns. However, the color scale is somewhat difficult to interpret. It may be helpful to adjust the color scale for better clarity or to include a color legend explaining the values represented."
  7. General (Line 90):
    • Current: "SoyBiPD" should be consistently written as "SoyBiPD" or "soyBiPD" throughout the manuscript.
    • Suggestion: Ensure consistent formatting for "SoyBiPD" throughout the manuscript (e.g., either capitalize or lowercase the gene name consistently).
  8. Minor Typo (Line 420):
    • Current: "in the absent of M. perniciosa"
    • Suggestion: "in the absence of M. perniciosa."
  9. Miscellaneous:
    • Throughout the manuscript, avoid unnecessary hyphenation in words like "pre-established" (should be "preestablished"), "protein-composition" (should be "protein composition"), and "fold-change" (should be "fold change").
      • Suggestion: Review the manuscript for unnecessary hyphenation and

Comments on the Quality of English Language

English needs to improve, especially by a subject specialist.

Author Response

Comment 1:
In the title, change the word "analyzes" to "analysis." Also, check the gene name for consistency. In the graphical abstract (Figure 10), the authors mention it as "SoyBIPD," but in many places throughout the manuscript, it is written as "BiP." Ensure that the gene name is correct and consistent throughout the paper.

Done

Comment 2:
Provide the affiliations for all authors, as they are missing in the paper.

Done

Comment 3:
The abstract is too long; it should typically be around 300-350 words. I suggest making it more concise and to the point, focusing only on the most important information.

We rewrite the abstract

Comment 4:
Figure 1: Italicize the scientific name of the tomato (Solanum lycopersicum).

Done

Comment 5:
Change the phrase "not inoculated" to "non-inoculated" throughout the manuscript.

Done

Comment 6:
I suggest supplementing Figure 1 with full plant images and adding a scale bar to the figure.

We decided to give a zoom in the plants in order to facilitate the symptom observation.

Comment 7:
Figures 2 and 3: Ensure that the figures are properly arranged and aligned. Additionally, the text in the figures is unclear and difficult to read. Increase the text size or make it bold to improve readability.

Legends from all pictures were reavaliated. The text size was the one from the program. We are not able to change it.

Comment 8:
Figure 4: Improve the figure for better visualization. The graph looks very simple, and the text is not easily readable. Enhancing the clarity would be beneficial.

Done

Comment 9:
Figure 7: Consider dividing the figure into two separate figures for better clarity and understanding.

Comment 10:
I suggest removing the rectangles around the figures throughout the manuscript, including Figures 1, 4, and 8.

Done

Comment 11:
Figure 9: Improve Figure 9 following the suggestions made for Figure 4.

Done

Comment 12:
Lines 127-128: If the paper is still under submission and not yet published, there is no need to include this statement. Please remove it.

Done

Comment 13:
In several places throughout the paper (e.g., lines 129-130), the authors have included material and method details in the results section. These should be moved to the Materials and Methods section.

Done

Comment 14:
The authors should provide abbreviations only at their first use and then use them consistently throughout the manuscript. For instance, "NT" is repeated many times, but the abbreviation is provided each time, which is unnecessary.

Done

Comment 15:
Result heading 2.2: The heading is too long. Typically, headings should be precise and concise. I suggest revising it to a more suitable, shorter version, and changing "UP-regulated" to "upregulated."

We rewrite the paper looking to make it more concise.

Comment 16:
Line 621: Change the heading to "Generation of SoyBiPD Transgenic Plants." Also, cross-check the gene name as mentioned in earlier comments for consistency.

Done

Comment 17:
In the Materials and Methods section, the authors divide sections into headings and subheadings. This is unnecessary; simply provide the main heading and explain the method. For example, "4.3 Proteomic Analysis" and "4.3.1 Total Protein Extraction" could be simplified to just "Total Protein Extraction."

Done

Comment 18:
Line 631: Change to "M. perniciosa inoculation."

Done

Comment 19:
Line 650: Change the heading to "Total Protein Extraction."

Done

Comment 20:
Line 786: Change the heading to "Hâ‚‚Oâ‚‚ Determination."

Done

Comment 21:
Line 860: Provide the number of replications as well.

In material and methods, line 795-797, we report the replicas number.

Comment 22: 

Line 627: Change the Molecular diagnosis to “Identification of the positive transgene was carried out…..”   

Done

Reviewer 4 Report

Comments and Suggestions for Authors

The authors present a study on the overexpression of the SoyBiPD gene in Solanum lycopersicum (tomato) and its effects on resistance against the phytopathogen Moniliophthora perniciosa, the causative agent of witches' broom disease in cocoa. Utilizing proteomic analysis, the authors identify key proteins and pathways involved in the enhanced defense mechanisms conferred by BiP overexpression. 

Here is some suggestions for authors,

1. The manuscript presents valuable and original research with significant potential impact in the field of plant pathology and biotechnology. However, to enhance clarity, methodological transparency, and the overall quality of the manuscript, the authors should address the identified weaknesses. Specifically, improvements in language clarity, detailed methodological descriptions, deeper discussion of results in the context of existing literature, and thorough proofreading are necessary before the manuscript can be considered for publication.

2. The methodology is generally thorough, more details on the criteria for protein identification and quantification thresholds would enhance reproducibility.

3. The description of the transformation process is adequate but could include more details on selection markers and transformation efficiency.

4. Correct instances such as "BiP-trangenic" to "BiP-transgenic," "trangenic" to "transgenic,".

Author Response

  1. The manuscript presents valuable and original research with significant potential impact in the field of plant pathology and biotechnology. However, to enhance clarity, methodological transparency, and the overall quality of the manuscript, the authors should address the identified weaknesses. Specifically, improvements in language clarity, detailed methodological descriptions, deeper discussion of results in the context of existing literature, and thorough proofreading are necessary before the manuscript can be considered for publication.

The language clarity was improved by authors and further corrected by a specialized English company.  Furthermore, the methodology as well as discussion was improved using the reviewer suggestions.

  1. The methodology is generally thorough, more details on the criteria for protein identification and quantification thresholds would enhance reproducibility.

Done

  1. The description of the transformation process is adequate but could include more details on selection markers and transformation efficiency.

Information included on line 614-615 and 617-618

  1. Correct instances such as "BiP-trangenic" to "BiP-transgenic," "trangenic" to "transgenic,".

Done

Reviewer 5 Report

Comments and Suggestions for Authors

The objective of this work is to study the defense mechanisms against Moniliophthora perniciosa, one of the main pathogens affecting the cocoa crop. 

The authors inoculated BiP plants (tomato superexpressing BiP) and control plants (Non-transgenic) with a mix of  M. perniciosa spores and performed a comparative proteomic analysis after 45 days after of inoculation. BiP, or Binding immunoglobulin protein, belongs to the HSP70 family (Heat Shock Proteins) and it is involved in many cell mechanisms, particularly protection against stress.
An appropriate bioinformatics analysis was performed on the proteomics data.

In addition to the aforementioned,  a western blot analysis, several enzymatic tests (including Guaiacol Peroxidase, Superoxide Dismutase, and β-1,3-Glucanase) as well as the determination of Hâ‚‚Oâ‚‚ levels were performed.

The research is very well carried out and with a relevant objective.

The identification of key proteins involved in the defense response, as well as the use of the BiP gene as a target, could lead to more resistant cultivars, impacting several agricultural crops, in addition to cocoa.

In general, the text is very well justified.  But it seems to me that some text editing needs to be done.

I  recommend accepting after minor revisions.

Below I attach my recommendations:

The abstract should be simplified a little

In general, the text is very well justified. But I would work a little more on the introduction to explain that tomatoes are used as a model to prevent a pest in cocoa.

Global cocoa production reached around 4,449 million tons in 2023, demonstrating the magnitude of this commodity. (add reference)

BiP is a molecular chaperone located in the lumen of the endoplasmic reticulum 85 (ER), belonging to the HSP70 family (Heat Shock Proteins). (add reference)

Author Response

The objective of this work is to study the defense mechanisms against Moniliophthora perniciosa, one of the main pathogens affecting the cocoa crop. 

The authors inoculated BiP plants (tomato superexpressing BiP) and control plants (Non-transgenic) with a mix of  M. perniciosa spores and performed a comparative proteomic analysis after 45 days after of inoculation. BiP, or Binding immunoglobulin protein, belongs to the HSP70 family (Heat Shock Proteins) and it is involved in many cell mechanisms, particularly protection against stress. 
An appropriate bioinformatics analysis was performed on the proteomics data.

In addition to the aforementioned,  a western blot analysis, several enzymatic tests (including Guaiacol Peroxidase, Superoxide Dismutase, and β-1,3-Glucanase) as well as the determination of Hâ‚‚Oâ‚‚ levels were performed. 

The research is very well carried out and with a relevant objective.

The identification of key proteins involved in the defense response, as well as the use of the BiP gene as a target, could lead to more resistant cultivars, impacting several agricultural crops, in addition to cocoa.

In general, the text is very well justified.  But it seems to me that some text editing needs to be done.

Done

Round 2

Reviewer 2 Report

Comments and Suggestions for Authors

Author addressed all the raises comments in revised version of the MS accordingly.